# Chlorpyrifos Acts as a Positive Modulator and an Agonist of *N*-Methyl-d-Aspartate (NMDA) Receptors: A Novel Mechanism of Chlorpyrifos-Induced Neurotoxicity

**DOI:** 10.3390/jox15010012

**Published:** 2025-01-16

**Authors:** Mahmoud Awad Sherif, Wayne G. Carter, Ian R. Mellor

**Affiliations:** 1School of Life Sciences, University of Nottingham, Nottingham NG7 2RD, UK; m_awad1985@mans.edu.eg; 2Department of Forensic Medicine and Clinical Toxicology, Faculty of Medicine, Mansoura University, Mansoura 35516, Egypt; 3Clinical Toxicology Research Group, School of Medicine, Royal Derby Hospital Centre, University of Nottingham, Derby DE22 3DT, UK; wayne.carter@nottingham.ac.uk

**Keywords:** chlorpyrifos, organophosphate, neurotoxicity, NMDA receptors, stem cells, *Xenopus* oocytes, two-electrode voltage clamp

## Abstract

Chlorpyrifos (CPF) is a broad-spectrum organophosphate insecticide. Long-term exposure to low levels of CPF is associated with neurodevelopmental and neurodegenerative disorders. The mechanisms leading to these effects are still not fully understood. Normal NMDA receptor (NMDAR) function is essential for neuronal development and higher brain functionality, while its inappropriate stimulation results in neurological deficits. Thus, the current study aimed to investigate the role of NMDARs in CPF-induced neurotoxicity. We show that NMDARs mediate CPF-induced excitotoxicity in differentiated human fetal cortical neuronal ReNcell CX stem cells. In addition, by using two-electrode voltage clamp electrophysiology of *Xenopus* oocytes expressing NMDARs, we show CPF potentiation of both GluN1-1a/GluN2A (EC_50_ ≈ 40 nM) and GluN1-1a/GluN2B (EC_50_ ≈ 55 nM) receptors, as well as reductions (approximately halved) in the NMDA EC_50_s and direct activation by 10 μM CPF of both receptor types. In silico molecular docking validated CPF’s association with NMDARs through relatively high affinity binding (−8.82 kcal/mol) to a modulator site at the GluN1–GluN2A interface of the ligand-binding domains.

## 1. Introduction

Organophosphorus insecticides have been among the most heavily used pesticides throughout the world for over half a century, with chlorpyrifos (diethyl 3,5,6-trichloro-2-pyridyl phosphonothioate) (CPF) representing one of the top-selling insecticides for many years [1,2]. Their primary target in insect pests is acetylcholinesterase (AChE), which is inhibited by CPF binding, resulting in the abnormal accumulation of acetylcholine in the synaptic cleft and persistent activation of acetylcholine receptors. This leads to paralysis of the insect pest [3,4].

Despite recent efforts to restrict its applications, such as bans in the USA, Europe, and the UK [5,6,7], CPF is still authorized for use in some parts of the world. In addition, long-term exposure to residual CPF in the environment, food, and from imported products is still of concern in areas where CPF use has already been restricted.

Based on epidemiological, in vivo, and in vitro studies, exposure to CPF has been associated with adverse effects on the developing nervous system [8,9]. For instance, an increased risk of attention deficit hyperactivity disorder (ADHD) has been reported in children who had increased urinary levels of CPF [10]. Prenatal exposure to CPF is associated with decreased intellectual development and overall IQ scores in seven-year-old children [11], and morphological and structural changes were detected in the developing brain using magnetic resonance imaging (MRI) [12]. Furthermore, the risk of autism has been reported to increase three-fold in children whose pregnant mothers lived within a mile of agricultural regions treated with CPF, when the mothers were exposed in their second trimester [13,14]. Neurobehavioral deficits have also been reported among agricultural CPF users in Egypt [15]. The mechanisms associated with these effects are not well understood.

Although insecticide formulations utilize active CPF, the parent compound can be metabolized by de-sulfuration to form chlorpyrifos-oxon (CPO), an even more potent AChE inhibitor than CPF [16,17]. This can be further hydrolyzed into less toxic metabolites such as 3,5,6-trichloro-2-pyridinol (TCPy) and diethyl phosphate (DEP) [17]. CPF is also directly oxidized via CYP450 enzymes to diethyl thiophosphate (DETP) and TCPy, which can be detected in urine and provide a means (in addition to blood) to quantify CPF exposure [18].

Repeated exposure to CPF has been associated with long-term neurotoxic effects at doses that are insufficient to induce cholinergic toxidrome [19]. Furthermore, numerous studies have suggested that CPF-induced neurotoxic effects are explained by AChE-independent mechanisms including induction of oxidative stress and adduction of secondary targets at exposure levels that are unlikely to cause sufficient or prolonged inhibition of AChE [20,21,22,23,24,25].

*N*-methyl-d-aspartate receptor (NMDAR) activation, resulting in excitotoxicity, is a possible contributory mechanism to CPF-induced neurotoxicity, through increased presynaptic glutamate release, inducing glutamate-mediated excitotoxicity in primary cortical culture [26] and enhanced corticostriatal glutamatergic neurotransmission in mice [27].

NMDARs are a subtype of ionotropic glutamate receptors that are widely distributed in the central nervous system (CNS) [28]. Normal NMDAR physiology is fundamental for neuronal development and higher brain functions including learning and memory [29], while its inappropriate or excessive activation causes excitotoxicity and subsequent neurodegeneration [30,31,32]. Thus, in the current study, we aimed to investigate the possible role of NMDARs in CPF-induced neurotoxicity. To test this hypothesis, we employed the human neural progenitor ReNcell CX cell line, which mimics normal in vivo brain development with differentiation into co-cultures of neurons, astrocytes, and oligodendrocytes [33] and the development of neuronal synapses [33,34]. In addition, the direct effect of CPF on NMDAR function was investigated by two-electrode voltage clamp (TEVC) electrophysiology using *Xenopus* oocytes expressing recombinant GluN1-1a/GluN2A or GluN1-1a/GluN2B subunits. The study then employed molecular docking to simulate the molecular interaction between CPF and the NMDAR and to predict the binding mode and affinity. These are all novel approaches to the study of CPF-induced neurotoxicity.

## 2. Materials and Methods

### 2.1. Chemicals and Reagents

Chlorpyrifos (*O*,*O*-diethyl *O*-(3,5,6-trichloropyridin-2-yl) phosphorothioate) PESTANAL, purity ≥ 98% was purchased from Sigma-Aldrich (Poole, UK). It was dissolved in DMSO to make stock solutions of 100 mM and kept at −20 °C. Working solutions were freshly prepared and final concentrations were prepared in culture media with the DMSO concentration not exceeding 0.1%. All other chemicals and reagents were purchased from Sigma-Aldrich (Poole, UK) unless otherwise specified.

### 2.2. Neuronal Stem Cell Culture and Differentiation

The human neural progenitor ReNcell CX cell line (Merck Millipore, Watford, UK) is a neural stem cell line derived from the developing human fetal brain cortex (ventral mesencephalic region) at 14 weeks of gestational age. The ReNcell CX cell line was expanded on laminin-coated flasks (20 μg/mL; Sigma-Aldrich, Poole, UK) in ReNcell Neural Stem Cell (NSC) Maintenance Medium (Merck Millipore, Watford, UK) containing fresh EGF (20 ng/mL; Merck Millipore, Watford, UK) and FGF-b (20 ng/mL; Merck Millipore, Watford, UK). Cultures were incubated at 37 °C in a 95% humidified atmosphere of 5% CO_2_. ReNcell CX neural progenitor cells were differentiated into a co-culture of neurons, astrocytes, and oligodendrocytes by replacing the complete ReNcell CX medium with EGF- and FGF-b-free medium; with the differentiation medium refreshed every 2 days throughout the 4-week differentiation period.

### 2.3. Cell Viability Assessment Using Live, Dead, and Apoptotic Cell Staining

A three-color fluorescence assay was performed as described by Kim et al. [35]. Non-fluorescent fluorescein diacetate (FDA) permeates into active and intact cells and is hydrolyzed by intracellular esterases into a highly fluorescent green fluorescein when illuminated with a fluorescein isothiocyanate (FITC; blue) excitation filter set (emission wavelength 460 nm). Propidium iodide (PI) is a nucleic acid dye that cannot permeate intact cell membranes and selectively stains the nuclei of dead cells or cells in late apoptotic stages. Stained nuclei omit red fluorescence under rhodamine (green) excitation (emission wavelength 650 nm). Live–dead cell assays followed the protocol described by Jones and Senft [36]. The cells were observed using a Nikon TS100 inverted microscope (Nikon Europe B.V., Amstelveen, The Netherlands) with incident fluorescence optics. Images were captured at 200× magnification and digitalized with a Moticam 2300 camera (Roper Scientific Inc., Duluth, GA, USA; resolution 3 MP).

Live/dead cell staining was analyzed quantitatively. The number of PI-stained nuclei was counted manually using the cell counting option in the ImageJ software (V2 Fiji-win64, University of Nottingham). The live cells were FDA stained, while cells that were categorized as being in apoptosis had a compromised membrane and were stained by a mixture of FDA and PI.

### 2.4. Expression of NMDARs in Xenopus Laevis Oocytes

Ovaries of mature female *Xenopus laevis* were supplied by the European *Xenopus* Resource Centre (University of Portsmouth, Portsmouth, UK). Upon receipt, oocytes were first separated with 1 mg/mL collagenase (Type 1A from *Clostridium histolyticum*; Sigma, Poole, UK) in Ca^2+^-free Barth’s gentamicin–theophylline–pyruvate (GTP) solution (96 mM NaCl, 2 mM KCl, 1 mM MgCl_2_, 5 mM HEPES, 2.5 mM pyruvic acid, 0.5 mM theophylline, 50 μg/mL gentamicin, pH 7.5) on a shaker for approximately 45 min at room temperature. Oocytes were then rinsed with Ca^2+^-free GTP solution six to eight times until a clear solution was obtained and incubated in normal Barth’s GTP solution (Ca^2+^-free GTP with inclusion of 1.8 mM CaCl_2_) at ~4 °C for at least half an hour prior to microinjection.

Healthy oocytes at developmental stages IV and V were injected with 50 nL of cRNA (into the cytoplasm) or plasmid DNA (into the nucleus) at a concentration of ~100 ng/μL using a Nanolitre 2010 Injector (World Precision Instruments, Hitchin, UK). cRNA or DNA plasmids encoding the rat NMDAR subunits GluN1-1a and GluN2A or GluN1-1a and GluN2B were mixed at ratios of 1:1 (by weight) for injection. The cRNA was synthesized from linearized plasmid DNA (pRK7 or pBluscript SK(−)) containing the GluN-encoding genes using an Invitrogen mMessage mMachine kit (Thermo Fisher Scientific, Loughborough, UK). Injected oocytes were kept in normal Barth’s GTP solution at 19 °C for two to three days for expression of NMDAR subunits before recordings.

### 2.5. TEVC Recordings

TEVC electrophysiology was used to assess the direct effects of CPF on NMDARs containing GluN1-1a and GluN2A or GluN1-1a and GluN2B subunits expressed in *Xenopus* oocytes. Pulled borosilicate glass capillaries (GC150TF-10, Harvard Apparatus, Cambridge, UK) (resistances 0.5–2 MΩ) were filled with 3 M KCl and used to impale injected oocytes for recording. Injected oocytes were continuously perfused with Mg^2+^-free *Xenopus* Ringer solution (95 mM NaCl, 2 mM KCl, 2 mM CaCl_2_, 5 mM HEPES, pH 7.5) at a flow rate of 5 mL/min, with test compounds applied using a Valvelink 8 gravity-fed perfusion system (Automate Scientific, Berkeley, CA, USA). Recordings were performed at 20–25 °C and oocytes were clamped at −75 mV holding potential in all experiments. Output current responses were transferred to a PC via an analogue-to-digital (A/D) converter (National Instruments PCI-6014, Austin, TX, USA) and recorded in WinEDR V3.2.7 software (John Dempster, Strathclyde Electrophysiology Software, Strathclyde Institute of Pharmacy and Biomedical Sciences, University of Strathclyde, Strathclyde, UK). Current changes in response to NMDA/glycine when applied with CPF were normalized to the control current without CPF.

### 2.6. Molecular Docking

The 3D crystal structure of unbound apo human GluN1/GluN2A ligand-binding domain (LBD) with a 1.81 Å resolution (PDBID; 5H8F) [37] was retrieved from the Protein Databank (https://www.rcsb.org/ (accessed on 10 January 2023)) and used for docking studies. The structure of CPF (PubChem CID: 2730) was retrieved from the PubChem database in SDF file format, which was then converted into PDBQT format just before docking. CPF (ligand) was docked with the target protein (GluN1/GluN2A LBD) using AutoDock Vina 1.5.6 as described by Mishra and Dey [38]. The receptor and ligand files were represented in PDBQT file format. For docking, the protein molecule was prepared by deleting heteroatoms and water followed by adding Kollman charges and polar hydrogen atoms using AutoDock tools 4.2.6. A grid box measuring 126 Å in each dimension was established, with a grid spacing of 0.4 Å, keeping the receptor rigid and the ligand as a flexible molecule. The 50 conformations of the molecules with binding energy and docking, the interaction energy of the ligand, its geometric coordinates, and a summary of the interaction energies, such as grid score, electrostatic energy, and van der Waals forces, were obtained using the Lamarckian genetic algorithm as the docking result. The intermolecular energy and other terms were calculated through the docking software. The ligand’s backbone and sidechain were flexible and allowed to dock with the receptor to form all possible conformations. After defining the binding site and receptor–ligand preparation, docking runs were launched from the command prompt. The interaction energy between the ligand and the receptor was calculated for the entire binding site and expressed as affinity (kcal/mol). Then the conformations were ranked based on the lowest energy obtained from the root mean square (RMS) deviation of each cluster. The binding interactions between the protein and ligands were further visualized and analyzed using PyMOL (version 3.0; https://www.pymol.org/ (accessed 24 July 2024)) and LigPlot software (version 2.2; https://www.ebi.ac.uk/thornton-srv/software/LigPlus/ (accessed 22 July 2024)).

### 2.7. Data Analysis

Results were expressed as means ± standard error of the mean (SEM). Statistical analysis was undertaken between concentrations using an independent samples Kruskal–Wallis test across cell culture conditions using IBM SPSS statistical software (version 26.0). A *p* < 0.05 indicated data were significantly different from the control. EC_50_ and maximum response values were obtained using non-linear regression by fitting a four-parametric logistic equation to concentration–response plots in GraphPad Prism 10 (GraphPad Software Inc., La Jolla, CA, USA):(1)$$Y=Min+\frac{Max-Min}{10^{\left(Log{EC}_{50}-X\right)S}+1}$$
where *Y* is the response to the agonist, *X* is the log_10_ concentration of the agonist, *EC*_50_ is the concentration of agonist that produces a half-maximum activation response, and *S* is the Hill slope. *EC*_50_*s* and maximum responses were compared for significant differences using an extra sum of squares F-test in GraphPad Prism 10.

## 3. Results

### 3.1. Ifenprodil Attenuates the CPF-Induced Decrease in Differentiated Neuronal Stem Cell Viability

Exposure of 4-week differentiated ReNcell CX cells to CPF at a concentration of 14 μM for 24 h triggered a significant increase in apoptosis (*p* < 0.001) and a significant decrease in cell viability from 96.4% to 71.4% (*p* < 0.001) (Figure 1A,B). Furthermore, significantly more (*p* < 0.0001) treated cells had transitioned to apoptosis (18.7%) compared to dead cells (3.91%) (Figure 1B). Co-administration of the non-competitive NMDAR antagonist, ifenprodil (IFN), at a concentration of 25 μM, significantly (*p* < 0.001) attenuated the CPF-induced decrease in ReNcell CX cell viability (Figure 1B); indicative that the CPF-induced decrease in cell viability is mediated by NMDARs.

### 3.2. CPF Potentiates NMDA/Gly-Evoked Currents

Co-application of CPF in increasing concentrations with NMDA and glycine potentiated the NMDA/glycine-evoked current in *Xenopus* oocytes expressing recombinant GluN1-1a/GluN2A and GluN1-1a/GluN2B receptors, when tested at −75 mV (Figure 2A,B). The CPF EC_50_s were ~40 nM and ~55 nM for GluN1-1a/GluN2A and GluN1-1a/GluN2B, respectively (Figure 2C).

CPF-induced potentiation of NMDA/glycine-evoked current was completely blocked by co-application of NMDAR antagonists, with GluN1-1a/GluN2A receptors blocked by 10 μM MK-801 (Figure 3A) and GluN1-1a/GluN2B receptors blocked by 100 μM IFN (Figure 3B).

### 3.3. CPF Reduced NMDA EC_50_s and Increased Its Maximal Response

The effect of CPF application on NMDA EC_50_ values was studied in *Xenopus* oocytes expressing GluN1-1a/GluN2A or GluN1-1a/GluN2B. Here, 10 μM glycine was present in all solutions. Co-application of 10 μM CPF with 0.1 to 1000 μM NMDA reduced the NMDA EC_50_ from 57.9 μM (95% CI 30.0–98.8) to 32.9 μM (95% CI 16.8–64.2) for GluN1-1a/GluN2A (Figure 4A) and from 70.0 μM (95% CI 39.9–111) to 35.5 μM (95% CI 20.7–59.3) for GluN1-1a/GluN2B (Figure 4B), although these reductions did not reach statistical significance (*p* = 0.255 and 0.075, respectively). The maximum responses significantly increased from 102% (95% CI 90.0–116) to 135% (95% CI 119–153) (*p* = 0.0043) for GluN1-1a/GluN2A (Figure 4A) and from 103% (95% CI 90.6–115) to 120% (95% CI 108–133) (*p* = 0.048) for GluN1-1a/GluN2B (Figure 4B).

### 3.4. CPF Can Activate NMDAR Alone or in Combination with Either NMDA or Glycine

CPF (10 μM) was directly applied to *Xenopus* oocytes expressing recombinant NMDAR subunits GluN1-1a/GluN2A or GluN1-1a/GluN2B at −75 mV (Figure 5). This resulted in NMDAR-mediated currents for both GluN1-1a/GluN2A (Figure 5A) and GluN1-1a/GluN2B (Figure 5B). CPF was also co-applied with either 10 μM glycine (at 1 μM) or 100 μM NMDA (at 10 μM) to *Xenopus* oocytes expressing recombinant NMDAR subunits GluN1-1a/GluN2A or GluN1-1a/GluN2B, respectively. CPF significantly (*p* ˂ 0.05) enhanced NMDAR-mediated current evoked by 10 μM glycine or 100 μM NMDA by ~1.7-fold and ~8.2-fold, respectively (Figure 5C–F).

### 3.5. CPF Interacts with NMDARs at a Positive Modulatory Site in the Interface Between the GluN1 and GluN2A Ligand-Binding Domains

CPF was docked into the GluN1/GluN2A ligand-binding domain (LBD) to assess the level of interaction and to consider the interacting residues between the protein and ligand complexes (Figure 6A–C). The CPF-GluN1/GluN2A LBD complex exhibited the lowest binding energy (highest affinity) of −8.82 kcal/mol through hydrogen bond interaction with Arg248(755) and hydrophobic interactions with Ile128(519), Pro141(532), Ser249(756), and Gly250(757) from the GluN1 subunit and hydrophobic interactions with Val128(526), Pro129(527), Phe130(528), Val131(529), Glu132(530), Leu263(780), and Val266(783) from the GluN2A subunit (numbers in parentheses are residue numbers in the full length subunits) (Figure 6B,C).

## 4. Discussion

CPF is an environmental hazard and a frequent contaminant of food produce [39,40]. Animal studies have suggested that exposure to low levels of CPF at key neurodevelopmental times can induce neurological deficits [19,41]. However, there is a need to further consider the potential for toxicological effects in human-based studies and models [42,43]. Therefore, the ReNcell CX (human) cell line was chosen for the current study and to comply with the principles of the 3Rs (reduction, refinement, and replacement in animal studies). A three-color fluorescence assay revealed that CPF-induced apoptosis was attenuated by the co-application of ifenprodil (IFN) in human cortical neuronal stem cells that were derived from ReNcell CX cells differentiated for 4 weeks. IFN is an allosteric non-competitive NMDAR antagonist with high selectivity for GluN2B-containing NMDARs [44,45]. This suggests that CPF-induced neurotoxicity can, at least in part, be mediated by (pathological) activation of GluN2B-containing NMDARs; a potentially novel mechanism of CPF-induced neurotoxicity. A significant increase in neuronal Ca^2+^ levels has been reported to mediate excitotoxic cell death induced by the pathological activation of NMDARs [46]. GluN1/GluN2B-containing NMDARs were reported to have slower kinetics of channel deactivation than GluN1/GluN2A subunits, allowing more Ca^2+^ influx [47]. Therefore, data obtained from the current study suggest that the differentiated cortical neuronal ReNcell CX cells express GluN2B-containing NMDARs at this developmental stage, consistent with previous studies reporting that expression of NMDARs containing GluN1-1a/GluN2B and GluN1-1a/GluN2D subunits is high in the pre- and early post-natal human brain [48,49]. Previous studies have also shown that differentiating neuronal stem cells derived from human fetal brain cortex can attain a time-dependent functional expression of NMDARs [50,51]. The modest level of apoptosis induced here by a relatively high concentration of CPF is likely due to the differentiation of ReNcell CX cells into various cell types [33], with only a subset of these being neurons expressing NMDARs.

A role for NMDARs in CPF-induced neurotoxicity was further confirmed using electrophysiological studies. TEVC recordings showed that CPF elicited NMDAR-mediated currents when applied directly in *Xenopus* oocytes expressing recombinant GluN1-1a/GluN2A or GluN1-1a/GluN2B NMDAR subunits. Furthermore, CPF potentiated NMDA/Gly-evoked currents in a concentration-dependent manner. Interestingly, the EC_50_ values for CPF potentiation are similar to or even lower than in vivo human exposure levels that have been reported. For instance, blood levels of CPF in some mothers (at delivery) or newborns living in an agricultural community (Salinas Valley, CA, USA) were as high as ≈4–5 µM, respectively [52]. In addition, a median concentration of 23.5 µM CPF was detected in meconium samples taken from newborns of 200 pregnant women who were at risk of exposure to CPF through dermal absorption, inhalation, and ingestion [53]. The NMDAR subunits GluN1-1a/GluN2A and GluN1-1a/GluN2B were specifically used in the current study as they represent two highly expressed NMDAR subunit combinations at different neurodevelopmental stages [54,55].

Molecular docking data confirmed that CPF could be best docked at the GluN1–GluN2A inter-dimer interface of the LBDs and with a relatively high binding affinity. A hydrogen bond can be formed between the nitrogen atom of the 3,5,6-trichloropyridine ring of CPF and Arg248(755) of the S2 domain of GluN1 and there are hydrophobic interactions between CPF and multiple residues of the S1 and S2 domains of GluN1 and GluN2A LBDs. All but two residues (Val131(529) → Ile and Val266(783) → Phe in GluN2B) that interact with CPF are shared between GluN1/GluN2A and GluN1/GluN2B subunits, and this could explain the relatively non-selective effect of CPF on both NMDAR subtypes in the current study. There is a paucity of data that consider NMDAR modulators that bind at the GluN1/GluN2 LBD interface. GNE-6901 and GNE-8324 are synthetic compounds that behave as positive allosteric modulators of NMDARs, which bind at the GluN1/GluN2A LBD dimer interface with selective action on GluN1/GluN2A NMDARs [37]. Interestingly, they bind at approximately the same location that we have predicted for CPF. The binding of CPF to this positive modulator site in the GluN1/GluN2A LBD interface may promote the formation of contacts between adjacent GluN1 and GluN2A LBDs, altering their normal configuration and leading to stabilization of an open channel state in the presence or absence of NMDA (or glutamate)/glycine binding. It would be expected that there will be two equivalent CPF binding sites per NMDAR, in which case, all four LBDs will be influenced by CPF binding.

The current study provides a novel mechanistic explanation of the potential impact of CPF exposure on the neurodevelopment and neurocognitive functions of the human brain, through pathological activation or potentiation by CPF of excessive NMDAR activation, since these receptors play an essential role in higher brain functions such as learning, memory, and synaptic plasticity [56,57]. Therefore, an increased prevalence of neurodevelopmental and neurodegenerative disorders in areas with high levels of exposure to CPF could correlate with CPF-induced neurotoxicity, including the disruption of normal NMDAR-mediated neuronal signaling and synaptic plasticity. Furthermore, CPF-mediated neurotoxicity, including the over-activation of NMDARs, could contribute to diminished function and the associated impact on IQ levels, neurodevelopmental delay, and neurological disorders such as ADHD, autism spectrum disorders, and deficits in cognitive function attributed to CPF exposures [8,12,15].

Interestingly, our TEVC data (Figure 2) provide evidence for CPF effects (in vitro) at concentrations below those reported for in vivo human exposure levels. Egyptian pesticide users occupationally exposed to CPF had an estimated internal exposure of 181 μg/kg/day (≈516 nM) [58], underlining the need for suitable personal protective equipment (PPE) during crop spraying. This exposure is higher than the no observed adverse effect level (NOAEL) of 100 μg/kg/day (≈285 nM) set by the European Food Safety Authority [59]. Hence, CPF-induced neurotoxicity may arise at exposure levels below those reported to cause effects on brain development in industry-funded risk assessment studies [60], highlighting the need to consider appropriate safety margins for exposures of the developing brain to xenobiotic environmental chemicals.

Although efforts are being implemented to improve testing paradigms for developmental neurotoxicity [61], there may be discrepancies between actual safety levels and those established by the chemical regulatory and environmental agencies, in particular for the consideration of the developmental neurotoxicity of chemicals. This can be attributed to limited developmental neurotoxicity testing in vivo [61,62] and the need to develop and implement a battery of appropriate developmental neurotoxicity tests [62,63,64], which may not always have been undertaken for chemical hazard risk assessment including that for neurotoxic pesticides. The need for developmental toxicity testing is particularly pertinent given the potential vulnerability of the developing nervous system to damage from CPF at relatively low exposure levels [12,65].

In summary, the current study has revealed that CPF can act as a positive modulator and activator of NMDARs, with a proposed high-affinity binding to a novel modulatory site as a contributory mechanism to CPF-induced neurotoxicity. Furthermore, the concentration range of CPF that induced toxicity was comparable (or lower) to that documented in human exposures. This highlights the need for a more comprehensive evaluation of CPF-induced developmental neurotoxicity by inclusion of sensitive methods such as electrophysiology and to consider safer pesticide alternatives.

## Figures and Tables

**Figure 1 jox-15-00012-f001:**
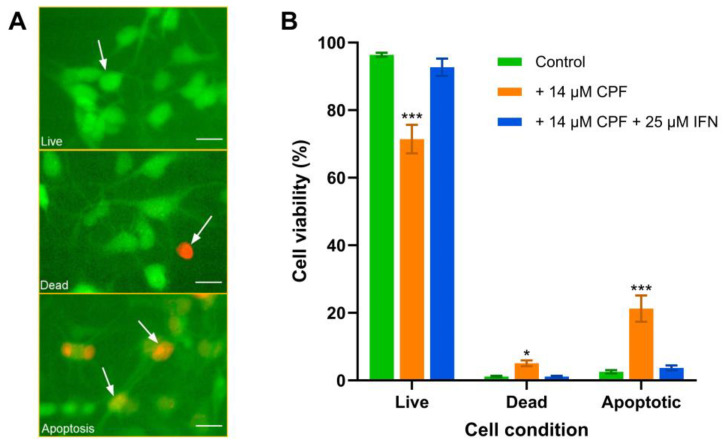
CPF-induced apoptosis and cell death of 4-week differentiated ReNcell CX cells. Cells were treated with 14 μM CPF, with or without 25 μM IFN, for 24 h and the percentage of live, dead, and apoptotic cells quantified using a three-color fluorescence assay. (**A**) Sample cell treatment showing live (evenly stained green with FDA), dead (stained red with PI), and apoptotic (mixture of FDA and PI) cells, highlighted with arrows. Scale bar is 20 µm. (**B**) Data points are means ± SEMs from two independent experiments in which triplicate wells were assessed with 6 captured images in each well from the control and treated cells. For marked significance: * *p* < 0.05 and *** *p* < 0.001 based on a Bonferroni post hoc correction test for significant differences from control.

**Figure 2 jox-15-00012-f002:**
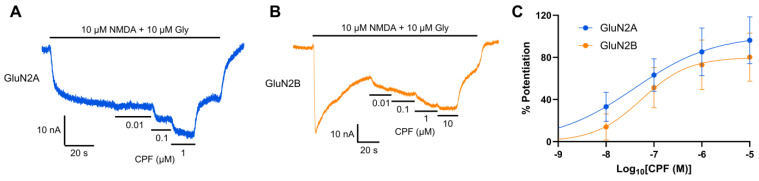
CPF potentiation of NMDA/glycine (10 μM/10 μM) responses in GluN1-1a/GluN2A and GluN1-1a/GluN2B-containing NMDARs. Sample TEVC recordings for CPF potentiation of current mediated by GluN1-1a/GluN2A (**A**) or GluN1-1a/GluN2B (**B**) at −75 mV. (**C**): Concentration–potentiation curves for CPF potentiated current mediated by GluN1-1a/GluN2A (n = 5) or GluN1-1a/GluN2B (n = 6). Percentage of potentiation (mean ± SEM) values were plotted against Log_10_ CPF concentration and fitted with Equation (1).

**Figure 3 jox-15-00012-f003:**
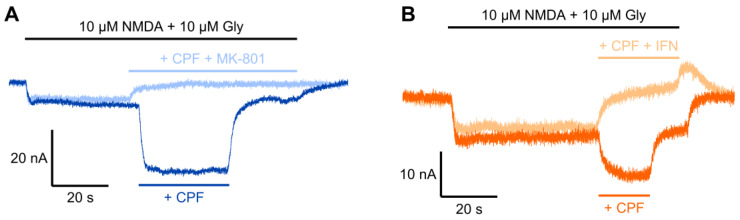
TEVC recordings showing antagonism of CPF-induced potentiation of NMDA/Gly-evoked currents in *Xenopus* oocytes expressing recombinant NMDARs. The 1 μM CPF potentiated NMDA/Gly-evoked current when co-applied with 10 μM NMDA and 10 μM Gly, was antagonized by co-application of 10 μM MK-801 or 100 μM IFN to *Xenopus* oocytes expressing GluN1-1a/GluN2A (**A**) or GluN1-1a/GluN2B (**B**), respectively. Recordings with or without antagonist were made from the same oocytes at −75 mV.

**Figure 4 jox-15-00012-f004:**
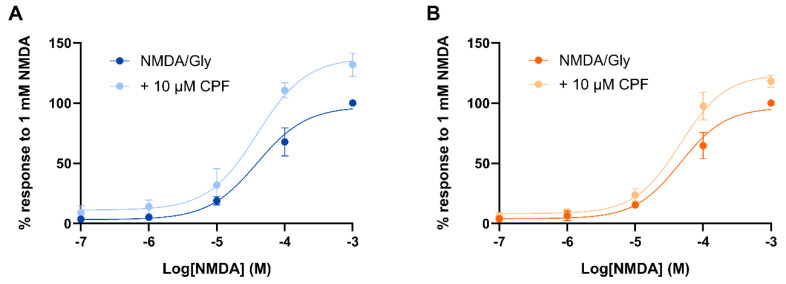
CPF reduced the NMDA EC_50_ and increased its maximum response. Concentration–response curves for NMDA-evoked (0.1 to 1000 µM; all with 10 μM Gly) current mediated by GluN1-1a/GluN2A (n = 5) (**A**) or GluN1-1a/GluN2B (n = 6) (**B**) in the absence and presence of 10 μM CPF. All data were expressed as a percentage of the response to 1 mM NMDA (mean ± SEM), plotted and fitted with Equation (1) to give EC_50_ and maximum response values.

**Figure 5 jox-15-00012-f005:**
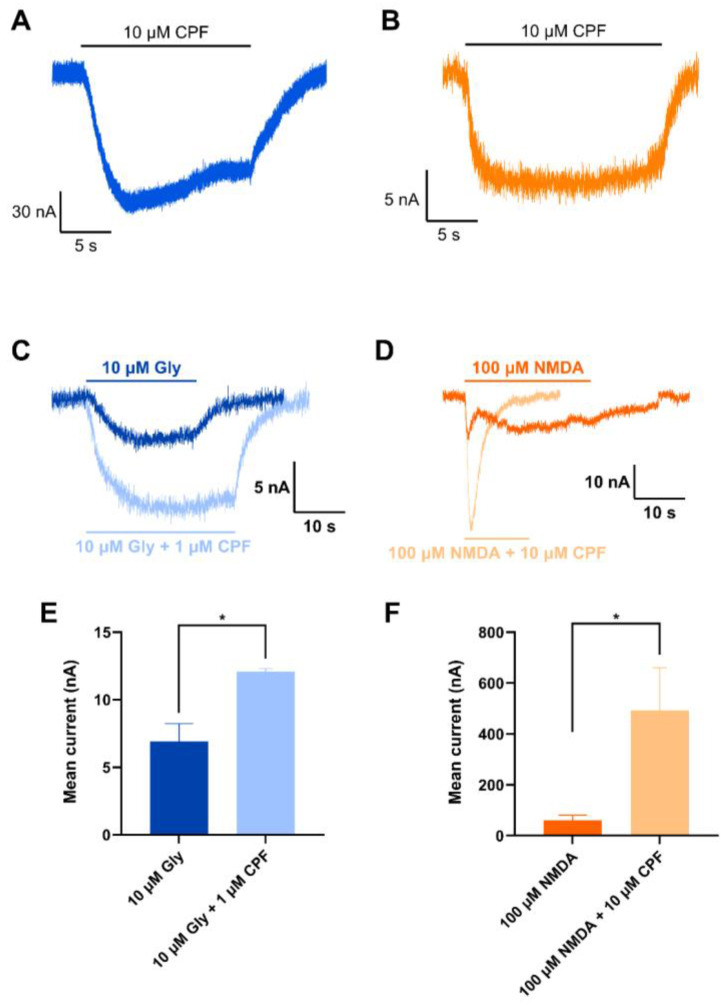
Effect of CPF alone or co-application of CPF with either NMDA or glycine in *Xenopus* oocytes expressing NMDARs. (**A**,**B**): TEVC recordings (at −75 mV) for direct application of 10 μM CPF to *Xenopus* oocytes expressing GluN1-1a/GluN2A (**A**) or GluN1-1a/GluN2B (**B**) showing elicited currents in both cases. (**C**,**D**): TEVC recordings for application of either 10 μM Gly ± 1 μM CPF or 100 μM NMDA ± 10 μM in *Xenopus* oocytes expressing GluN1-1a/GluN2A (**C**) or GluN1-1a/GluN2B (**D**), respectively. (**E**,**F**): Currents were significantly enhanced for glycine at GluN1-1a/GluN2A (n = 3) (**E**) and for NMDA at GluN1-1a/GluN2B (n = 5) (**F**). Data shown are means ± SEM of the peak current response. Statistical analysis was performed using an unpaired Student’s *t*-test (two-tailed). For marked significance: * indicates changes that were significantly different from NMDA- or glycine-only-evoked responses with *p* ˂ 0.05.

**Figure 6 jox-15-00012-f006:**
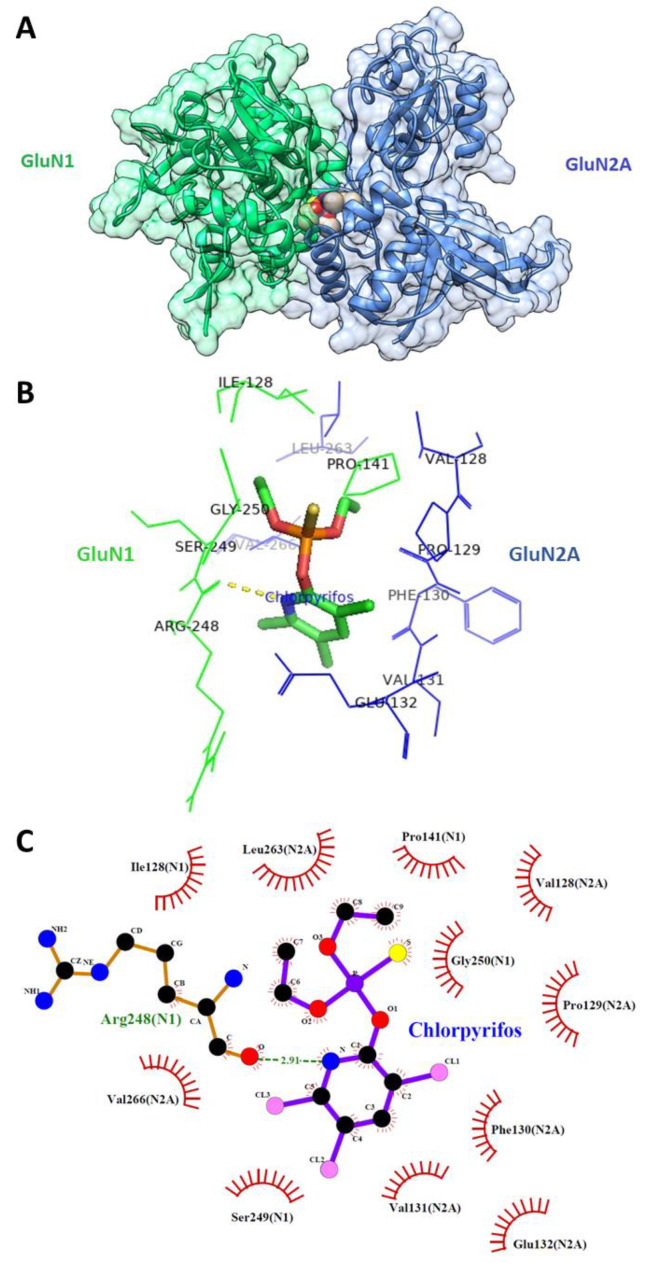
Three-dimensional (**A**,**B**) and two-dimensional (**C**) poses showing interaction of CPF with the GluN1/GluN2A LBD interface. CPF forms a hydrogen bond (dashed line) with Arg248(755) and hydrophobic interactions with Ile128(519), Pro141(532), Ser249(756), and Gly250(757) from the GluN1 subunit LBD and hydrophobic interactions with Val128(526), Pro129(527), Phe130(528), Val131(529), Glu132(530), Leu263(780), and Val266(783) from the GluN2A subunit LBD (numbers in parentheses are residue numbers in the full length subunits).

## Data Availability

The original contributions presented in this study are included in the article. Further inquiries can be directed to the corresponding author.

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
