# Peer review of "Chlorpyrifos Acts as a Positive Modulator and an Agonist of N-Methyl-d-Aspartate (NMDA) Receptors: A Novel Mechanism of Chlorpyrifos-Induced Neurotoxicity"

_jox, 2025, doi:10.3390/jox15010012_

Round 1
Reviewer 1 Report
Comments and Suggestions for Authors
The manuscript focused on CPF induced neurotoxicity through NMDAR receptor. Both the experimental and molecular docking has been performed to support the results. Overall the manuscript has been written in quite concise way and can be accepted for publication after minor revisions.
In introduction, docking can be introduced including if such study has been done before or not.
The author can try to improve discussion part as the reference to existing literature is missing.
EC50 needs to be discussed properly in the manuscript as it is very important for regulatory aspects.
Author Response
Comment 1: In introduction, docking can be introduced including if such study has been done before or not.
Response 1: Docking was introduced as an approach used in this study (last sentence of introduction) but there are no previous studies considering CPF and NMDAR to discuss further. We have clarified that all of the studies here have been conducted for the first time in line 83.
Comment 2: The author can try to improve discussion part as the reference to existing literature is missing.
Response 2: There are 34 literature citations in the discussion section and most of these are first introduced in the discussion. We don’t think we have missed any that are relevant but would be happy to be guided by the referee if we are wrong.
Comment 3: EC50 needs to be discussed properly in the manuscript as it is very important for regulatory aspects.
Response 3: The EC50 values for CPF potentiation of NMDARs were reported in the manuscript (section 3.2 and Figure 2) and further discussed in comparison to concentrations detected in vivo (second paragraph of the discussion).
Reviewer 2 Report
Comments and Suggestions for Authors
In this manuscript, Sherif et al. have evaluated the activity of Chlorpyrifos (CPF), an organophosphate insecticide inhibiting acetylcholine esterase, on NMDA receptors. They observe that CPF behaves as a mixed aginist/co-agonist on NMDA receptors. CPF extensive use raises public concern as it could be associated with neurodevelopmental issues. The manuscript is clear and well-written and the conclusions straightforward.
Here are my comments:
1-The neurotoxicity experiments are, to some extent, rather unconvivncing. Indeed, firstly, why was a signle dose of CPF tested and not a full concentration-dependent study performed ? Secondly, the concentration used for toxicity experiments (14µM) produces rather modest effect (20 % cell death) on cell survival. This is rather strinking when comparing to the EC50 of CPF for NMDA receptors reported here which is around 50nM. This needs to be discussed. In addition, the combination NMDA+CPF should probably be tested for its neurotoxic action (as it has been for TEVC functional studies).
2-With regards to the data shown in the figures 5, an antagonist of the NMDA Glycine binding site should be tested to confirm the co-agonust action of CPF on NMDA receptors. In addition, why were the combination of Gly+CPF and NMDA+CPF (Figures 5C and 5D resepctively) applied longer than Gly or NMDA alone ?
Author Response
Comment 1: The neurotoxicity experiments are, to some extent, rather unconvivncing. Indeed, firstly, why was a signle dose of CPF tested and not a full concentration-dependent study performed?
Response 1: This was just to clearly demonstrate the toxic effect of CPF on cultures containing neurons, while the focus of the manuscript is on characterising the effect of CPF on NMDARs as a potential contributing mechanism. We are not in a position to do further experiments in the small timeframe available for revision.
Comment 2: Secondly, the concentration used for toxicity experiments (14µM) produces rather modest effect (20 % cell death) on cell survival. This is rather strinking when comparing to the EC50 of CPF for NMDA receptors reported here which is around 50nM. This needs to be discussed.
Response 2: This modest effect likely arises from the fact that the cultures of differentiated ReNcell CX cells consist of several cell types, with only a portion of these expressing NMDARs. So, 20% cell death could translate to 100% neuronal cell death. We have added a comment in the discussion (lines 312-315). It may also be limited by the fact that this is not a functioning system and likely only represents activation of NMDARs by CPF alone rather than potentiation of agonist/co-agonist effects; that EC50 value is for potentiation of NMDA/Gly evoked currents in oocytes, so may not be relevant in this context.
Comment 3: In addition, the combination NMDA+CPF should probably be tested for its neurotoxic action (as it has been for TEVC functional studies).
Response 3: We would also need to do NMDA/Gly alone. We are not in a position to do further experiments within the required response time. These additional experiments would not be expected to have an impact on the outcome of the study.
Comment 4: With regards to the data shown in the figures 5, an antagonist of the NMDA Glycine binding site should be tested to confirm the co-agonust action of CPF on NMDA receptors.
Response 4: As mentioned above, we are not in a position to do further experiments at present. Again, we would need to do additional experiments to assess the effect of a glutamate site antagonist as well. We considered that these would not tell us much, if anything, more than the experiments with NMDA + CPF or glycine + CPF, demonstrating that occupying the CPF site plus either one of the glutamate or glycine sites is sufficient to activate the NMDARs.
Comment 5: In addition, why were the combination of Gly+CPF and NMDA+CPF (Figures 5C and 5D resepctively) applied longer than Gly or NMDA alone?
Response 5: Applications were manually controlled and generally continued for a minimum of approximately 15 s but longer if a response had not reached equilibrium. This did not affect the response maxima measured.